# TWIN NETWORKS: MATCHING THE FUTURE FOR SEQUENCE GENERATION

**Dmitriy Serdyuk,**[*][♦] **Nan Rosemary Ke,**[*][♦][‡] **Alessandro Sordoni**[♥]
**Adam Trischler,**[♥] **Chris Pal**[♣][♦] **& Yoshua Bengio**[¶][♦]

[♦] Montreal Institute for Learning Algorithms (MILA), Canada
[♥] Microsoft Research, Canada
[♣] Ecole Polytechnique, Canada
[¶] CIFAR Senior Fellow
[‡] Work done at Microsoft Research
[*] **Authors contributed equally**
`serdyuk@iro.umontreal.ca, rosemary.nan.ke@gmail.com`

## ABSTRACT

We propose a simple technique for encouraging generative RNNs to plan ahead. We train a "backward" recurrent network to generate a given sequence in reverse order, and we encourage states of the forward model to predict cotemporal states of the backward model. The backward network is used only during training, and plays no role during sampling or inference. We hypothesize that our approach eases modeling of long-term dependencies by implicitly forcing the forward states to hold information about the longer-term future (as contained in the backward states). We show empirically that our approach achieves 9% relative improvement for a speech recognition task, and achieves significant improvement on a COCO caption generation task.

## 1 INTRODUCTION

Recurrent Neural Networks (RNNs) are the basis of state-of-art models for generating sequential data such as text and speech. RNNs are trained to generate sequences by predicting one output at a time given all previous ones, and excel at the task through their capacity to remember past information well beyond classical $n$-gram models (Bengio et al., 1994; Hochreiter & Schmidhuber, 1997). More recently, RNNs have also found success when applied to conditional generation tasks such as speech-to-text (Chorowski et al., 2015; Chan et al., 2016), image captioning (Xu et al., 2015) and machine translation (Sutskever et al., 2014; Bahdanau et al., 2014).

RNNs are usually trained by *teacher forcing*: at each point in a given sequence, the RNN is optimized to predict the next token given all preceding tokens. This corresponds to optimizing one-step-ahead prediction. As there is no explicit bias toward planning in the training objective, the model may prefer to focus on the most recent tokens instead of capturing subtle long-term dependencies that could contribute to global coherence. Local correlations are usually stronger than long-term dependencies and thus end up dominating the learning signal. The consequence is that samples from RNNs tend to exhibit local coherence but lack meaningful global structure. This difficulty in capturing long-term dependencies has been noted and discussed in several seminal works (Hochreiter, 1991; Bengio et al., 1994; Hochreiter & Schmidhuber, 1997; Pascanu et al., 2013).

Recent efforts to address this problem have involved augmenting RNNs with external memory (Dieng et al., 2016; Grave et al., 2016; Gulcehre et al., 2017a), with unitary or hierarchical architectures (Arjovsky et al., 2016; Serban et al., 2017), or with explicit planning mechanisms (Gulcehre et al., 2017b). Parallel efforts aim to prevent overfitting on strong local correlations by regularizing the states of the network, by applying dropout or penalizing various statistics (Moon et al., 2015; Zaremba et al., 2014; Gal & Ghahramani, 2016; Krueger et al., 2016; Merity et al., 2017).

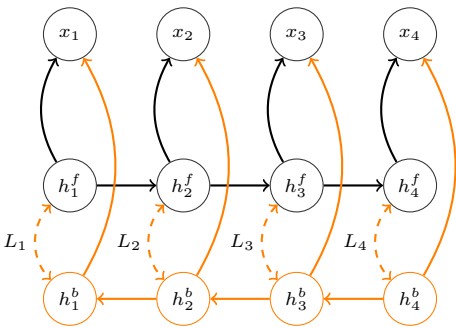

Figure 1: The forward and the backward networks predict the sequence $s = \{x_1, ..., x_4\}$ independently. The penalty matches the forward (or a parametric function of the forward) and the backward hidden states. The forward network receives the gradient signal from the log-likelihood objective as well as $L_t$ between states that predict the same token. The backward network is trained only by maximizing the data log-likelihood. During the evaluation part of the network colored with orange is discarded. The cost $L_t$ is either a Euclidean distance or a learned metric $||g(h_t^f) - h_t^b||_2$ with an affine transformation $g$. Best viewed in color.

In this paper, we propose *TwinNet*,[1] a simple method for regularizing a recurrent neural network that encourages modeling those aspects of the past that are predictive of the long-term future. Succinctly, this is achieved as follows: in parallel to the standard forward RNN, we run a "twin" backward RNN (with no parameter sharing) that predicts the sequence in reverse, and we encourage the hidden state of the forward network to be close to that of the backward network used to predict the same token. Intuitively, this forces the forward network to focus on the past information that is useful to predicting a specific token and that is *also* present in and useful to the backward network, coming from the future (Fig. 1).

In practice, our model introduces a regularization term to the training loss. This is distinct from other regularization methods that act on the hidden states either by injecting noise (Krueger et al., 2016) or by penalizing their norm (Krueger & Memisevic, 2015; Merity et al., 2017), because we formulate explicit auxiliary targets for the forward hidden states: namely, the backward hidden states. The activation regularizer (AR) proposed by Merity et al. (2017), which penalizes the norm of the hidden states, is equivalent to the TwinNet approach with the backward states set to zero. Overall, our model is driven by the intuition (a) that the backward hidden states contain a summary of the future of the sequence, and (b) that in order to predict the future more accurately, the model will have to form a better representation of the past. We demonstrate the effectiveness of the TwinNet approach experimentally, through several conditional and unconditional generation tasks that include speech recognition, image captioning, language modelling, and sequential image generation. To summarize, the contributions of this work are as follows:

- We introduce a simple method for training generative recurrent networks that regularizes the hidden states of the network to anticipate future states (see Section 2);

- The paper provides extensive evaluation of the proposed model on multiple tasks and concludes that it helps training and regularization for conditioned generation (speech recognition, image captioning) and for the unconditioned case (sequential MNIST, language modelling, see Section 4);

- For deeper analysis we visualize the introduced cost and observe that it negatively correlates with the word frequency (more surprising words have higher cost).

---

[1] The source code is available at https://github.com/dmitriy-serdyuk/twin-net/.

## 2 MODEL

Given a dataset of sequences $\mathcal{S} = \{s^1, \ldots, s^n\}$, where each $s^k = \{x_1, \ldots, x_{T_k}\}$ is an observed sequence of inputs $x_i \in \mathcal{X}$, we wish to estimate a density $p(s)$ by maximizing the log-likelihood of the observed data $\mathcal{L} = \sum_{i=1}^{n} \log p(s^i)$. Using the chain rule, the joint probability over a sequence $x_1, \ldots, x_T$ decomposes as:

$$p(x_1, \ldots, x_T) = p(x_1)p(x_2|x_1)... = \prod_{t=1}^{T} p(x_t|x_1, \ldots, x_{t-1}). \tag{1}$$

This particular decomposition of the joint probability has been widely used in language modeling (Bengio et al., 2003; Mikolov, 2010) and speech recognition (Bahl et al., 1983). A recurrent neural network is a powerful architecture for approximating this conditional probability. At each step, the RNN updates a hidden state $h_t^f$, which iteratively summarizes the inputs seen up to time $t$:

$$h_t^f = \Phi_f(x_{t-1}, h_{t-1}^f), \tag{2}$$

where $f$ symbolizes that the network reads the sequence in the forward direction, and $\Phi_f$ is typically a non-linear function, such as a LSTM cell (Hochreiter & Schmidhuber, 1997) or a GRU (Cho et al., 2014). Thus, $h_t^f$ forms a representation summarizing information about the sequence's past. The prediction of the next symbol $x_t$ is performed using another non-linear transformation on top of $h_t^f$, i.e. $p_f(x_t|x_{<t}) = \Psi_f(h_t^f)$, which is typically a linear or affine transformation (followed by a softmax when $x_t$ is a symbol). The basic idea of our approach is to encourage $h_t^f$ to contain information that is useful to predict $x_t$ and which is also compatible with the upcoming (future) inputs in the sequence. To achieve this, we run a twin recurrent network that predicts the sequence in reverse and further require the hidden states of the forward and the backward networks to be close. The backward network updates its hidden state according to:

$$h_t^b = \Phi_b(x_{t+1}, h_{t+1}^b), \tag{3}$$

and predicts $p_b(x_t|x_{>t}) = \Psi_b(h_t^b)$ using information only about the future of the sequence. Thus, $h_t^f$ and $h_t^b$ both contain useful information for predicting $x_t$, coming respectively from the past and future. Our idea consists in penalizing the distance between forward and backward hidden states leading to the same prediction. For this we use the Euclidean distance (see Fig. 1):

$$L_t(s) = \|g(h_t^f) - h_t^b\|_2, \tag{4}$$

where the dependence on $x$ is implicit in the definition of $h_t^f$ and $h_t^b$. The function $g$ adds further capacity to the model and comes from the class of parameterized affine transformations. Note that this class includes the identity tranformation. As we will show experimentally in Section 4, a learned affine transformation gives more flexibility to the model and leads to better results. This relaxes the strict match between forward and backward states, requiring just that the forward hidden states are predictive of the backward hidden states.[2]

The total objective maximized by our model for a sequence $s$ is a weighted sum of the forward and backward log-likelihoods minus the penalty term, computed at each time-step:

$$\mathcal{F}(s) = \sum_t \log p_f(x_t|x_{<t}) + \log p_b(x_t|x_{>t}) - \alpha \, L_t(s), \tag{5}$$

where $\alpha$ is an hyper-parameter controlling the importance of the penalty term. In order to provide a more stable learning signal to the forward network, we only propagate the gradient of the penalty term through the forward network. That is, we avoid co-adaptation of the backward and forward networks. During sampling and evaluation, we discard the backward network.

The proposed method can be easily extended to the conditional generation case. The forward hidden-state transition is modified to

$$h_t^f = \Phi_f\left(x_{t-1}, \left[h_{t-1}^f, c\right]\right), \tag{6}$$

where $c$ denotes the task-dependent conditioning information, and similarly for the backward RNN.

---

[2] Matching hidden states is equivalent to matching joint distributions factorized in two different ways, since a given state contains a representation of all previous states for generation of all later states and outputs. For comparison, we made several experiments matching outputs of the forward and backward networks rather than their hidden states, which is equivalent to matching $p(x_t|x_{<t})$ and $p(x_t|x_{>t})$ separately for every $t$. None of these experiments converged.

## 3 RELATED WORK

Bidirectional neural networks (Schuster & Paliwal, 1997) have been used as powerful feature extractors for sequence tasks. The hidden state at each time step includes both information from the past and the future. For this reason, they usually act as better feature extractors than the unidirectional counterpart and have been successfully used in a myriad of tasks, e.g. in machine translation (Bahdanau et al., 2015), question answering (Chen et al., 2017) and sequence labeling (Ma & Hovy, 2016). However, it is not straightforward to apply these models to sequence generation (Zhang et al., 2018) due to the fact that the ancestral sampling process is not allowed to look into the future. In this paper, the backward model is used to regularize the hidden states of the forward model and thus is only used during training. Both inference and sampling are strictly equivalent to the unidirectional case.

Gated architectures such as LSTMs (Hochreiter & Schmidhuber, 1997) and GRUs (Chung et al., 2014) have been successful in easing the modeling of long term-dependencies: the gates indicate time-steps for which the network is allowed to keep new information in the memory or forget stored information. Graves et al. (2014); Dieng et al. (2016); Grave et al. (2016) effectively augment the memory of the network by means of an external memory. Another solution for capturing long-term dependencies and avoiding gradient vanishing problems is equipping existing architectures with a hierarchical structure (Serban et al., 2017). Other works tackled the vanishing gradient problem by making the recurrent dynamics unitary (Arjovsky et al., 2016). In parallel, inspired by recent advances in "learning to plan" for reinforcement learning (Silver et al., 2016; Tamar et al., 2016), recent efforts try to augment RNNs with an explicit planning mechanism (Gulcehre et al., 2017b) to force the network to commit to a plan while generating, or to make hidden states predictive of the far future (Li et al., 2017).

Regularization methods such as noise injection are also useful to shape the learning dynamics and overcome local correlations to take over the learning process. One of the most popular methods for neural network regularization is dropout (Srivastava et al., 2014). Dropout in RNNs has been proposed in (Moon et al., 2015), and was later extended in (Semeniuta et al., 2016; Gal & Ghahramani, 2016), where recurrent connections are dropped at random. Zoneout (Krueger et al., 2016) modifies the hidden state to regularize the network by effectively creating an ensemble of different length recurrent networks. Krueger & Memisevic (2015) introduce a "norm stabilization" regularization term that ensures that the consecutive hidden states of an RNN have similar Euclidean norm. Recently, Merity et al. (2017) proposed a set of regularization methods that achieve state-of-the-art on the Penn Treebank language modeling dataset. Other RNN regularization methods include the weight noise (Graves, 2011), gradient clipping (Pascanu et al., 2013) and gradient noise (Neelakantan et al., 2015).

## 4 EXPERIMENTAL SETUP AND RESULTS

We now present experiments on conditional and unconditional sequence generation, and analyze the results in an effort to understand the performance gains of TwinNet. First, we examine conditional generation tasks such as speech recognition and image captioning, where the results show clear improvements over the baseline and other regularization methods. Next, we explore unconditional language generation, where we find our model does not significantly improve on the baseline. Finally, to further determine what tasks the model is well-suited to, we analyze a sequential imputation task, where we can vary the task from unconditional to strongly conditional.

### 4.1 SPEECH RECOGNITION

We evaluated our approach on the conditional generation for character-level speech recognition, where the model is trained to convert the speech audio signal to the sequence of characters. The forward and backward RNNs are trained as conditional generative models with soft-attention (Chorowski et al., 2015). The context information $c$ is an encoding of the audio sequence and the output sequence $s$ is the corresponding character sequence. We evaluate our model on the Wall Street Journal (WSJ) dataset closely following the setting described in Bahdanau et al. (2016). We use 40 mel-filter bank features with delta and delta-deltas with their energies as the acoustic in-

Table 1: Average character error rate (CER, %) on WSJ dataset decoded with the beam size 10. We compare the attention model for speech recognition ("Baseline," Bahdanau et al., 2016); the regularizer proposed by Krueger & Memisevic (2015) ("Stabilizing norm"); penalty on the L2 norm of the forward states (Merity et al., 2017) ("AR"), which is equivalent to TwinNet when all the hidden states of the backward network are set to zero. We report the results of our model ("TwinNet") both with $g = I$, the identity mapping, and with a learned $g$.

| Model | Test CER | Valid CER |
|---|---|---|
| Baseline | 6.8 | 9.0 |
| Baseline + Gaussian noise | 6.9 | 9.1 |
| Baseline + Stabilizing Norm | 6.6 | 9.0 |
| Baseline + AR | 6.5 | 8.9 |
| Baseline + TwinNet ($g = I$) | 6.6 | 8.7 |
| Baseline + TwinNet (learnt $g$) | **6.2** | **8.4** |

puts to the model, these features are generated according to the Kaldi s5 recipe (Povey et al., 2011). The resulting input feature dimension is 123.

We observe the Character Error Rate (CER) for our validation set, and we early stop on the best CER observed so far. We report CER for both our validation and test sets. For all our models and the baseline, we follow the setup in Bahdanau et al. (2016) and pretrain the model for 1 epoch, within this period, the context window is only allowed to move forward. We then perform 10 epochs of training, where the context window looks freely along the time axis of the encoded sequence, we also perform annealing on the models with 2 different learning rates and 3 epochs for each annealing stage. We use the AdaDelta optimizer for training. We perform a small hyper-parameter search on the weight $\alpha$ of our twin loss, $\alpha \in \{2.0, 1.5, 1.0, 0.5, 0.25, 0.1\}$, and select the best one according to the CER on the validation set.[3]

**Results**   We summarize our findings in Table 1. Our best performing model shows relative improvement of 12% comparing to the baseline. We found that the TwinNet with a learned metric (learnt $g$) is more effective than strictly matching forward and hidden states. In order to gain insights on whether the empirical usefulness comes from using a backward recurrent network, we propose two ablation tests. For "Gaussian Noise," the backward states are randomly sampled from a Gaussian distribution, therefore the forward states are trained to predict white noise. For "AR," the backward states are set to zero, which is equivalent to penalizing the norm of the forward hidden states (Merity et al., 2017). Finally, we compare the model with the "Stabilizing Norm" regularizer (Krueger & Memisevic, 2015), that penalizes the difference of the norm of consecutive forward hidden states. Results shows that the information included in the backward states is indeed useful for obtaining a significant improvement.

**Analysis**   The training/validation curve comparison for the baseline and our network is presented in Figure 2a.[4] The TwinNet converges faster than the baseline and generalizes better. The L2 cost raises in the beginning as the forward and backward network start to learn independently. Later, due to the pressure of this cost, networks produce more aligned hidden representations. Figure 3 provides examples of utterances with L2 plotted along the time axis. We observe that the high entropy words produce spikes in the loss for such words as "uzi." This is the case for rare words which are hard to predict from the acoustic information. To elaborate on this, we plot the L2 cost averaged over a word depending on the word frequency. The average distance decreases with the increasing frequency. The histogram comparison (Figure 2b) for the cost of rare and frequent words reveal that the not only the average cost is lower for frequent words, but the variance is higher for rare words. Additionally, we plot the dependency of the L2 cost cross-entropy cost of the forward network (Figure 2c) to show that the conditioning also plays the role in the entropy of the output, the losses are not absolutely correlated.

---

[3]The best hyperparameter was 1.5.

[4]The saw tooth pattern of both training curves corresponds to shuffling within each epoch as was previously noted by Bottou (2009).

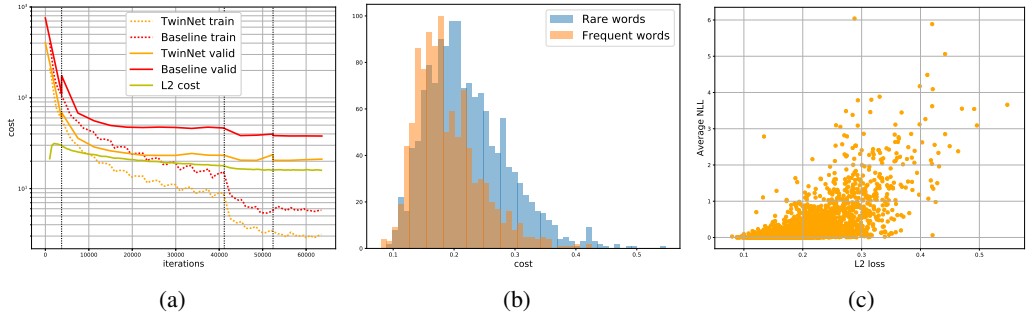

|     |     |     |
| --- | --- | --- |
| (a) | (b) | (c) |

Figure 2: Analysis for speech recognition experiments. **(a)**: Training curves comparison for Twin-Nets and the baseline network. Dotted vertical lines denote stages of pre-training, training, and two stages of annealing. The L2 cost is plotted alongside. The TwinNet converges to a better solution as well as provides better generalization. **(b)**: Comparison of histograms of the cost for rare words (first 1500) versus frequent words (all other). The cost is averaged over characters of a word. The distribution of rare words is wider and tends to produce higher L2 cost. **(c)**: L2 loss vs. average cross-entropy loss.

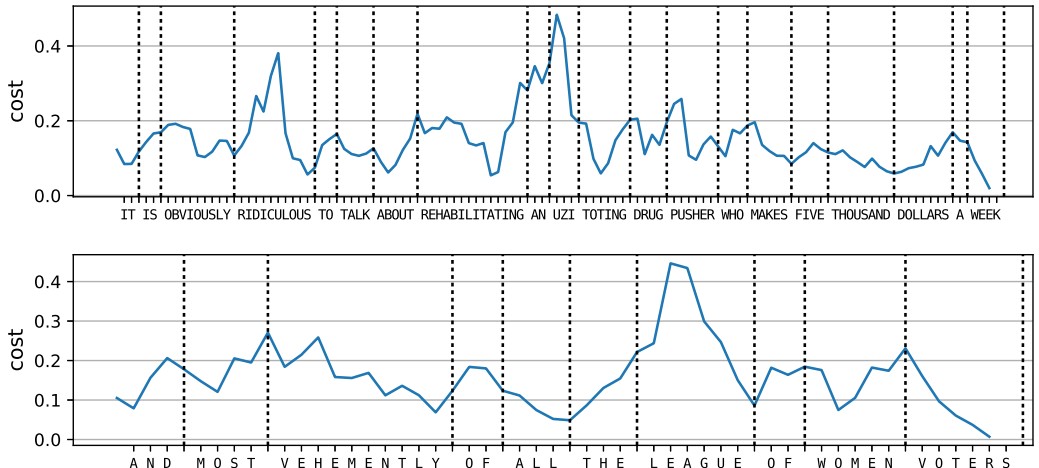

Figure 3: Example of the L2 loss plotted along the time axis. Notice that spikes correspond to rare words given the acoustic information where the entropy of the prediction is high. Dotted vertical lines are plotted at word boundary positions.

## 4.2 IMAGE CAPTIONING

We evaluate our model on the conditional generation task of image captioning task on Microsoft COCO dataset (Lin et al., 2014). The MS COCO dataset covers 82,783 training images and 40,504 images for validation. Due to the lack of standardized split of training, validation and test data, we follow Karpathy's split (Karpathy & Fei-Fei, 2015; Xu et al., 2015; Wang et al., 2016). These are 80,000 training images and 5,000 images for validation and test. We do early stopping based on the validation CIDEr scores and we report BLEU-1 to BLEU-4, CIDEr, and Meteor scores. To evaluate the consistency of our method, we tested TwinNet on both encoder-decoder ('Show&Tell', Vinyals et al., 2015) and soft attention ('Show, Attend and Tell', Xu et al., 2015) image captioning models.[5]

We use a Resnet (He et al., 2016) with 101 and 152 layers pre-trained on ImageNet for image classification. The last layer of the Resned is used to extract 2048 dimensional input features for the attention model (Xu et al., 2015). We use an LSTM with 512 hidden units for both "Show & Tell" and soft attention. Both models are trained with the Adam (Kingma & Ba, 2014) optimizer with a

---

[5] Following the setup in `https://github.com/ruotianluo/neuraltalk2.pytorch`.

Table 2: Results for image captioning on the MS COCO dataset, the higher the better for all metrics (BLEU 1 to 4, METEOR, and CIDEr). We reimplement both Show&Tell (Vinyals et al., 2015) and Soft Attention (Xu et al., 2015) in order to add the twin cost. We use two types of images features extracted either with Resnet-101 or Resnet-152.

| Models | B-1 | B-2 | B-3 | B-4 | METEOR | CIDEr |
|---|---|---|---|---|---|---|
| DeepVS (Karpathy & Fei-Fei, 2015) | 62.5 | 45.0 | 32.1 | 23.0 | 19.5 | 66.0 |
| ATT-FCN (You et al., 2016) | 70.9 | 53.7 | 40.2 | 30.4 | 24.3 | - |
| Show & Tell (Vinyals et al., 2015) | - | - | - | 27.7 | 23.7 | 85.5 |
| Soft Attention (Xu et al., 2015) | 70.7 | 49.2 | 34.4 | 24.3 | 23.9 | - |
| Hard Attention (Xu et al., 2015) | 71.8 | 50.4 | 35.7 | 25.0 | 23.0 | - |
| MSM (Yao et al., 2016) | 73.0 | 56.5 | 42.9 | 32.5 | 25.1 | 98.6 |
| Adaptive Attention (Lu et al., 2017) | **74.2** | **58.0** | **43.9** | **33.2** | **26.6** | **108.5** |
| *No attention, Resnet101* | | | | | | |
| Show&Tell (Our impl.) | 69.4 | 51.6 | 36.9 | 26.3 | 23.4 | 84.3 |
| + TwinNet | **71.8** | **54.5** | **39.4** | **28.0** | **24.0** | **87.7** |
| *Attention, Resnet101* | | | | | | |
| Soft Attention (Our impl.) | 71.0 | 53.7 | 39.0 | 28.1 | 24.0 | 89.2 |
| + TwinNet | **72.8** | **55.7** | **41.0** | **29.7** | **25.2** | **96.2** |
| *No attention, Resnet152* | | | | | | |
| Show&Tell (Our impl.) | 71.7 | 54.4 | 39.7 | 28.8 | 24.8 | 93.0 |
| + TwinNet | **72.3** | **55.2** | **40.4** | **29.3** | **25.1** | **94.7** |
| *Attention, Resnet152* | | | | | | |
| Soft Attention (Our impl.) | 73.2 | 56.3 | 41.4 | 30.1 | **25.3** | 96.6 |
| + TwinNet | **73.8** | **56.9** | **42.0** | **30.6** | 25.2 | **97.3** |

Table 3: **(left)** Test set negative log-likelihood for binarized sequential MNIST, where ▾ denotes lower performance of our model with respect to the baselines. **(right)** Perplexity results on WikiText-2 and Penn Treebank (Merity et al., 2017). AWD-LSTM refers to the model of (Merity et al., 2017) trained with the official implementation at http://github.com/salesforce/awd-lstm/.

| Model | MNIST |
|---|---|
| DBN 2hl (Germain et al., 2015) | $\approx$84.55 |
| NADE (Uria et al., 2016) | 88.33 |
| EoNADE-5 2hl (Raiko et al., 2014) | 84.68 |
| DLGM 8 (Salimans et al., 2014) | $\approx$85.51 |
| DARN 1hl (Gregor et al., 2015) | $\approx$84.13 |
| DRAW (Gregor et al., 2015) | $\leq$80.97 |
| P-Forcing$_{(3\text{-layer})}$ (Lamb et al., 2016) | 79.58 |
| PixelRNN$_{(1\text{-layer})}$ (Oord et al., 2016b) | 80.75 |
| PixelRNN$_{(7\text{-layer})}$ (Oord et al., 2016b) | 79.20 |
| PixelVAE (Gulrajani et al., 2016) | 79.02▾ |
| MatNets (Bachman, 2016) | 78.50▾ |
| Baseline LSTM$_{(3\text{-layers})}$ | 79.87 |
| + TwinNet$_{(3\text{-layers})}$ | **79.35** |
| Baseline LSTM$_{(3\text{-layers})}$ + dropout | 79.59 |
| + TwinNet$_{(3\text{-layers})}$ | **79.12** |

| Penn Treebank | Valid | Test |
|---|---|---|
| LSTM (Zaremba et al., 2014) | 82.2 | 78.4 |
| 4-layer LSTM (Melis et al., 2017) | 67.9 | 65.4 |
| 5-layer RHN (Melis et al., 2017) | 64.8 | 62.2 |
| AWD-LSTM | 61.2 | 58.8 |
| + TwinNet | **61.0** | **58.3** |

| WikiText-2 | Valid | Test |
|---|---|---|
| 5-layer RHN (Melis et al., 2017) | 78.1 | 75.6 |
| 1-layer LSTM (Melis et al., 2017) | 69.3 | 65.9 |
| 2-layer LSTM (Melis et al., 2017) | 69.1 | 65.9 |
| AWD-LSTM | 68.7 | 65.8 |
| + TwinNet | **68.0** | **64.9** |

learning rate of $10^{-4}$. TwinNet showed consistent improvements over "Show & Tell" (Table 2). For the soft attention model we observe small but consistent improvements for majority of scores.

## 4.3 UNCONDITIONAL GENERATION: SEQUENTIAL MNIST AND LANGUAGE MODELING

We investigate the performance of our model in pixel-by-pixel generation for sequential MNIST. We follow the setting described by Lamb et al. (2016): we use an LSTM with 3-layers of 512 hidden

units for both forward and backward LSTMs, batch size 20, learning rate 0.001 and clip the gradient norms to 5. We use Adam (Kingma & Ba, 2014) as our optimization algorithm and we decay the learning rate by half after $5, 10$, and $15$ epochs. Our results are reported at the Table 3 (left). Our baseline LSTM implementation achieves 79.87 nats on the test set. We observe that by adding the TwinNet regularization cost consistently improves performance in this setting by about 0.52 nats. Adding dropout to the baseline LSTM is beneficial. Further gains were observed by adding both dropout and the TwinNet regularization cost. This last model achieves 79.12 nats on test set. Note that this result is competitive with deeper models such as PixelRNN (Oord et al., 2016b) (7-layers) and PixelVAE (Gulrajani et al., 2016) which uses an autoregressive decoder coupled with a deep stochastic auto-encoder.

As a last experiment, we report results obtained on a language modelling task using the PennTree Bank and WikiText-2 datasets (Merity et al., 2017). We augment the state-of-the-art AWD-LSTM model (Merity et al., 2017) with the proposed TwinNet regularization cost. The results are reported in Table 3 (right).

## 5    DISCUSSION

In this paper, we presented a simple recurrent neural network model that has two separate networks running in opposite directions during training. Our model is motivated by the fact that states of the forward model should be predictive of the entire future sequence. This may be hard to obtain by optimizing one-step ahead predictions. The backward path is discarded during the sampling and evaluation process, which makes the sampling process efficient. Empirical results show that the proposed method performs well on conditional generation for several tasks. The analysis reveals an interpretable behaviour of the proposed loss.

One of the shortcomings of the proposed approach is that the training process doubles the computation needed for the baseline (due to the backward network training). However, since the backward network is discarded during sampling, the sampling or inference process has the exact same computation steps as the baseline. This makes our approach applicable to models that requires expensive sampling steps, such as PixelRNNs (Oord et al., 2016b) and WaveNet (Oord et al., 2016a). One of future work directions is to test whether it could help in conditional speech synthesis using WaveNet.

We observed that the proposed approach yield minor improvements when applied to language modelling with PennTree bank. We hypothesize that this may be linked to the amount of entropy of the target distribution. In these high-entropy cases, at any time-step in the sequence, the distribution of backward states may be highly multi-modal (many possible futures may be equally likely for the same past). One way of overcoming this problem would be to replace the proposed L2 loss (which implicitly assumes a unimodal distribution of the backward states) by a more expressive loss obtained by either employing an inference network (Kingma & Welling, 2013) or distribution matching techniques (Goodfellow et al., 2014). We leave that for future investigation.

## ACKNOWLEDGMENTS

The authors would like to acknowledge the support of the following agencies for research funding and computing support: NSERC, Calcul Québec, Compute Canada, the Canada Research Chairs, CIFAR, and Samsung. We would also like to thank the developers of Theano Theano Development Team (2016), Blocks and Fuel van Merriënboer et al. (2015), and Pytorch for developments of great frameworks. We thank Aaron Courville, Sandeep Subramanian, Marc-Alexandre Côté, Anirudh Goyal, Alex Lamb, Philemon Brakel, Devon Hjelm, Kyle Kastner, Olivier Breuleux, Phil Bachman, and Gaétan Marceau Caron for useful feedback and discussions.

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
