# OpenReview forum: "Twin Networks: Matching the Future for Sequence Generation"
_ICLR.cc/2018/Conference — Accept (Poster)_

### Official Review · AnonReviewer1 · 2017-11-25
**The paper reads well, has sufficient reference. The idea is simple and well explained. Positive empirial results support the proposed regularizer.**

**Rating:** 8
**Confidence:** 4

**Review:**

Twin Networks: Using the Future as a Regularizer

** PAPER SUMMARY **

The authors propose to regularize RNN for sequence prediction by forcing states of the main forward RNN to match the state of a secondary backward RNN. Both RNNs are trained jointly and only the forward model is used at test time. Experiments on conditional generation (speech recognition, image captioning), and unconditional generation (MNIST pixel RNN, language models) show the effectiveness of the regularizer.

** REVIEW SUMMARY **

The paper reads well, has sufficient reference. The idea is simple and well explained. Positive empirial results support the proposed regularizer.

** DETAILED REVIEW **

Overall, this is a good paper. I have a few suggestions along the text but nothing major.

In related work, I would cite co-training approaches. In effect, you have two view of a point in time, its past and its future and you force these two views to agree, see  (Blum and Mitchell, 1998) or Xu, Chang, Dacheng Tao, and Chao Xu. "A survey on multi-view learning." arXiv preprint arXiv:1304.5634 (2013). I would also relate your work to distillation/model compression which tries to get one network to behave like another. On that point, is it important to train the forward and backward network jointly or could the backward network be pre-trained?

In section 2, it is not obvious to me that the regularizer (4) would not be ignored in absence of regularization on the output matrix. I mean, the regularizer could push h^b to small norm, compensating with higher norm for the output word embeddings. Could you comment why this would not happen?

In Section 4.2, you need to refer to Table 2 in the text. You also need to define the evaluation metrics used. In this section, why are you not reporting the results from the original Show&Tell paper? How does your implementation compare to the original work?

On unconditional generation, your hypothesis on uncertainty is interesting and could be tested. You could inject uncertainty in the captioning task for instance, e.g. consider that multiple version of each word e.g. dogA, dogB, docC which are alternatively used instead of dog with predefined substitution rates. Would your regularizer still be helpful there? At which point would it break?

---

> ### Author Response · Authors · 2018-01-05
> **Response to Reviewer 1**
>
> We thank the reviewer for your positive feedback and comments!
>
> Q: “On that point, is it important to train the forward and backward network jointly or could the backward network be pre-trained?”
>
> A: As the gradient of the regularization term is not backpropagated through the backward network, the backward model can indeed be pre-trained.
>
> Q: In section 2, it is not obvious to me that the regularizer (4) would not be ignored in absence of regularization on the output matrix. I mean, the regularizer could push h^b to small norm, compensating with higher norm for the output word embeddings. Could you comment why this would not happen?
>
> A: The L2 cost to match the forward and backward states (4) is not backpropagated to the backward model, i.e. the hidden states of the backward h^b are not optimized with respect to the twin cost. Therefore, the backward hidden states may be pushed to a small norm only if it’s beneficial for the reverse language modeling objective.
>
> Q: In this section, why are you not reporting the results from the original Show&Tell paper? How does your implementation compare to the original work?
>
> A: The original ShowTell uses the Inception v3 network for the feature extraction. Therefore the performance is not comparable to our baseline trained with features extracted with ResNet. This result is added to the table now.
>
> Q: dogA, dogB, docC which are alternatively used instead of dog with predefined substitution rates. Would your regularizer still be helpful there? At which point would it break?
> A: The experiment on multiple versions of each word is indeed very interesting, thanks for the suggestion. Our new results suggest that the method works also in unconditioned case. Please refer to the comment to all reviewers.

---

### Official Review · AnonReviewer3 · 2017-11-26
**Review (Post-Rebutal)**

**Rating:** 6
**Confidence:** 4

**Review:**


1) Summary
This paper proposes a recurrent neural network (RNN) training formulation for encouraging RNN the hidden representations to contain information useful for predicting future timesteps reliably. The authors propose to train a forward and backward RNN in parallel. The forward RNN predicts forward in time and the backward RNN predicts backwards in time. While the forward RNN is trained to predict the next timestep, its hidden representation is forced to be similar to the representation of the backward RNN in the same optimization step. In experiments, it is shown that the proposed method improves training speed in terms of number of training iterations, achieves 0.8 CIDEr points improvement over baselines using the proposed training, and also achieves improved performance for the task of speech recognition.


2) Pros:
+ Novel idea that makes sense for learning a more robust representation for predicting the future and prevent only local temporal correlations learned.
+ Informative analysis for clearly identifying the strengths of the proposed method and where it is failing to perform as expected.
+ Improved performance in speech recognition task.
+ The idea is clearly explained and well motivated.


3) Cons:
Image captioning experiment:
In the experimental section, there is an image captioning result in which the proposed method is used on top of two baselines. This experiment shows improvement over such baselines, however, the performance is still worse compared against baselines such as Lu et al, 2017 and Yao et al, 2016. It would be optimal if the authors can use their training method on such baselines and show improved performance, or explain why this cannot be done.


Unconditioned generation experiments:
In these experiments, sequential pixel-by-pixel MNIST generation is performed in which the proposed method did not help. Because of this, two conditioned set ups are performed: 1) 25% of pixels are given before generation, and 2) 75% of pixels are given before generation. The proposed method performs similar to the baseline in the 25% case, and better than the baseline in the 75% case. For completeness, and to come to a stronger conclusion on how much uncertainty really affects the proposed method, this experiment needs a case in which 50% of the pixels are given. Observing 25% of the pixels gives almost no information about the identity of the digit and it makes sense that it’s hard to encode the future, however, 50% of the pixels give a good idea of what the digit identity is. If the authors believe that the 50% case is not necessary, please feel free to explain why.


Additional comments:
The method is shown to converge faster compared to the baselines, however, it is possible that the baseline may finish training faster (the authors do acknowledge the additional computation needed in the backward RNN).
It would be informative for the research community to see the relationship of training time (how long it takes in hours) versus how fast it learns (iterations taken to learn).

Experiments on RL planning tasks would be interesting to see (Maybe on a simple/predictable environment).


4) Conclusion
The paper proposes a method for training RNN architectures to better model the future in its internal state supervised by another RNN modeling the future in reverse. Correctly modeling the future is very important for tasks that require making decisions of what to do in the future based on what we predict from the past. The proposed method presents a possible way of better modeling the future, however, some the results do not clearly back up the claim yet. The given score will improve if the authors are able to address the stated issues.


POST REBUTTAL RESPONSE:
The authors have addressed the comments on the MNIST experiments and show better results, however, as far as I can see, they did not address my concern about the comparisons on the image captioning experiment. In the image captioning experiment the authors choose two networks (Show & Tell and Soft attention) that they improve using the proposed method that end up performing similar to the second best baseline (Yao et al. 2016) based on Table 3 and their response. I requested for the authors to use their method on the best performing baselines (i.e. Yao et al. 2016 or Liu et al. 2017) or explain why this cannot be done (maybe my request was not clearly stated). Applying the proposed method on the strong baselines would highlight the author's claims more strongly than just applying on the average performing chosen baselines. This request was not addressed and instead the authors just improved the average performing baselines in Table 3 to meet the best baselines. Given, that the authors were able to improve the results in the sequential MNIST and improve the average baselines, my rating improves one point. However, I still have concerns about this method not being shown to improve the best methods presented in Table 3 which would give a more solid result. My rating changes to marginally above threshold for acceptance.

---

> ### Author Response · Authors · 2018-01-05
> **Response to Reviewer 3**
>
> We thank the reviewer for the feedback and comments.
>
> Q: “3) Cons: Image captioning experiment:
> In the experimental section, there is an image captioning result in which the proposed method is used on top of two baselines. ”
>
> A: We acknowledge this, and we have significant improvements on our image captioning experiments, we have the following improvements, which we also summarized in the comments to all reviewers.
> We run SAT models with Resnet 152 features. SAT with Twinnet achieved similar performance to (Yao et al, 2016). SAT with Twinnet vs Yao et al, 2016 performances are B1: 73.8 (vs 73.0 Yao), B2: 56.9 (vs 56.5 Yao), B3: 42.0 (vs 42.9 Yao), B4: 30.6 (vs 32.5 Yao), Meteor: 25.2 (vs 25.1 Yao), Cider: 97.3 (vs 98.6 Yao)
> We have significant improvements on ST and SAT with Resnet 101 features compared to the baseline.
>
> Q: “Observing 25% of the pixels gives almost no information about the identity of the digit and it makes sense that it’s hard to encode the future, however, 50% of the pixels give a good idea of what the digit identity is. If the authors believe that the 50% case is not necessary, please feel free to explain why.”
>
> A: We have run more thorough examinations, with larger regularization hyperparameter values, for the unconditioned generation. We now have consistent improvements on both conditioned and unconditioned generation tasks. Please see our comment to all reviewers.
>
> Q: “It would be informative for the research community to see the relationship of training time (how long it takes in hours) versus how fast it learns (iterations taken to learn).”
>
> A: We are currently running the forward and backward RNN consecutively, therefore training Twinnet takes around twice the amount of time as training the forward alone. We measured the batch time for running the SAT baseline on Resnet152 feature takes 0.181s/minibatch, and SAT with Twinet is 0.378s/minibatch. Both experiments are run on TitanXP GPUs. We also measured convergence. For the ASR task, the convergence is the same in terms of number of epochs as it can be seen in the learning curve in the paper. For the image captioning, some TwinNet models converge faster, while others have similar convergence rate similar compared to the baseline.
>
>
> Q: “Experiments on RL planning tasks would be interesting to see (Maybe on a simple/predictable environment).”
>
> A: This is a very nice idea! We could see this being used for model-based RL (planning). However, we feel that this tasks deserves to be a separate paper on its own. It would require some amount of investigation to understand how forward and backward would interact in a RL setting. It would indeed be very interesting future work to see how Twinnet could be used in this setting.

---

> > ### Author Response · Authors · 2018-01-14
> > **Re: Review**
> >
> > We’d like to thank you again for your review and feedbacks! We have updated our paper with your suggestions (including significantly improved Image Captioning results, comparisons to the Yao et al 2016. model and training time). With more thorough experimentation, we have also shown that Twinnet works for unconditioned generation (as well as conditioned generation).
> >
> > Would you have any other questions regarding the rebuttal, especially regards to the image captioning experiments and unconditioned generation?

---

### Official Review · AnonReviewer2 · 2017-11-27
**Simple way to regularize recurrent sequence generators, limited applicability.**

**Rating:** 7
**Confidence:** 4

**Review:**

** post-rebuttal revision **

I thank the authors for running the baseline experiments, especially for running the TwinNet to learn an agreement between two RNNs going forward in time. This raises my confidence that what is reported is better than mere distillation of an ensemble of rnns. I am raising the score.

** original review **


The paper presents a way to regularize a sequence generator by making the hidden states also predict the hidden states of an RNN working backward.

Applied to sequence-to-sequence networks, the approach requires training one encoder, and two separate decoders, that generate the target sequence in forward and reversed orders. A penalty term is added that forces an agreement between the hidden states of the two decoders. During model evaluation only the forward decoder is used, with the backward operating decoder discarded. The method can be interpreted to generalize other recurrent network regularizers, such as putting an L2 loss on the hidden states.

Experiments indicate that the  approach is most successful when the regularized RNNs are conditional generators, which emit sequences of low entropy, such as decoders of a seq2seq speech recognition network. Negative results were reported when the proposed regularization technique was applied to language models, whose output distribution has more entropy.

The proposed regularization is evaluated with positive results on a speech recognition task and on an  image captioning task, and with negative results (no improvement, but also no deterioration) on a language modeling and sequential MNIST digit generation tasks.

I have one question about baselines: is the proposed approach better than training to forward generators and force an agreement between them (in the spirit of the concurrent ICLR submission https://openreview.net/forum?id=rkr1UDeC-)?

Also, would using the backward RNN, e.g. for rescoring, bring another advantage? In other words, what is (and is there) a gap between an ensemble of a forward and backward rnn and the forward-rnn only, but trained with the state-matching penalty?

Quality:
The proposed approach is well motivated and the experiments show the limits of applicability range of the technique.

Clarity:
The paper is clearly written.

Originality:
The presented idea seems novel.

Significance:
The method may prove to be useful to regularize recurrent networks, however I would like to see a comparison with ensemble methods. Also, as the authors note the method seems to be limited to conditional sequence generators.

Pros and cons:
Pros: the method is simple to implement, the paper lists for what kind of datasets it can be used.
Cons: the method needs to be compared with typical ensembles of models going only forward in time, it may turn that it using the backward RNN is not necessary

---

> ### Author Response · Authors · 2017-12-11
> **Clarification**
>
> We thank you for your review, feedback and suggestions.
>
> We would like to clarify your question about ensembles. Unlike ensemble methods, our method does not average over multiple predictive models at the evaluation time. Forward and backward networks are “coupled” during training and the backward predictions are discarded during testing. Would you like to see a comparison between ensembles of baselines compared to ensembles of TwinNets?

---

> ### Author Response · Authors · 2018-01-05
> **Response to Reviewer 2**
>
> We thank the reviewer for the feedback and comments. We now have improved results across all tasks (image captioning, sequential MNIST and language modelling), we included those in a brief summary in our comment that was addressed to all reviewers.
>
> Q: “I have one question about baselines: is the proposed approach better than training to forward generators and force an agreement between them, …”
>
> A: We run the following experiment to test out this hypothesis. We run 2 separate forward generators and force an agreement on those 2 generators. We tested out this model on WSJ dataset for speech recognition, and the model has a validation CER of 9.1% ( vs 9.0% for Baseline) and 8.4% for TwinNet. We conclude that TwinNet works better compared to running 2 forward models and force an agreement between them.
>
> Q: “Also, would using the backward RNN, e.g. for rescoring, bring another advantage? ...”
>
> A: This is an interesting question to think about the backward RNN doing rescoring. Although, we are not totally clear on what ensembles of forward model would be in this case, and we have not received a clarification on this based on our earlier comments.
>
> We took a best guess of what the reviewer meant in this case. We assume that the reviewer means that we run multiple forward generators and use the backward generator to rescore the results of the forward generator and hence picking the best out of the forward generators. In general, this is an interesting idea, although it is different enough from our current idea that it would deserve credit to be a separate paper on its own.
>
> Two things to note for the idea of running multiple forward generator and using the backward to rescore is
> Inference is at least twice as expensive compared to our current setup, which does not use the backward model during test time.
>
> As there are multiple choices for rescoring function, it is not trivial to decide which rescoring function to use. In some cases, the rescoring function could also be non-differentiable (for example WER or CER for speech recognition, or Bleu score for image captioning or machine translation).
>
> Q: “Cons: the method needs to be compared with typical ensembles of models going only forward in time, it may turn that it using the backward RNN is not necessary”
>
> A: We have indeed run the experiment with multiple (2 in our case) forward models and force an agreement between them, and we did not see improvements over the baseline. This supports our hypothesis that the backward RNN is indeed useful and necessary.

---

### Author Response · Authors · 2017-12-11
**Thank reviewers**

We thank all the reviewers for your feedback and suggestions, it was very helpful and allowed us to gain more insights. We plan to run the requested experiments and update the experimental results in the upcoming weeks.

---

### Author Response · Authors · 2018-01-05
**General comment to all reviewers.**

More thorough experimentation allowed us to significantly improve Twinnet results for image captioning, sequential MNIST, and language modelling tasks. We summarize our new results here and update the paper.

- Image Captioning: We have significantly improved results for ShowTell (ST) and ShowAttendTell (SAT) models with TwinNet (Table 2)
    1. ResNet 101 features + ST + Twinnet
        Improvement by more than 3 Cider points
        B1: 71.8 (vs 69.,4),  B2: 54.5 (vs 51.6), B3: 39.4 (vs 36.9), B4: 28.0 (vs 26.3), Meteor: 24.0 (vs 23.4), Cider: 87.7 (vs 84.3)
    2. ResNet 101 features + SAT + Twinnet
        Improvement by more than 5 Cider points
        B1: 72.8 (vs 71.0), B2:  55.7 (vs 53.0),  B3: 41.0(vs 39.0), B4: 29.7 (vs 28.1), Meteor: 25.2 (vs 25.0), Cider: 96.2 (vs 89.2)
    3. ResNet 152 features + SAT + Twinnet
        Improvement by 0.7 Cider points
        B1: 73.8 (vs 73.2), B2:  56.9 (vs 56.3),  B3: 42.0 (vs 41.4), B4: 30.6 (vs 30.1), Meteor: 25.2 (vs 25.3), Cider: 97.3 (vs 96.6)
-  Sequential MNIST (Table 3 (left))
    1. LSTM with Dropout  + Twinnet has NLL of 79.12 (vs 79.59 for Baseline)
    2. LSTM + TwinNet has NLL of 79.35 (vs 79.87 for Baseline)
-  Wikitext 2 (Table 3 (right))
    - AWD + LSTM + TwinNet has valid perplexity: 68.0 (vs 68.7) and Test perplexity: 64.9 (vs 65.8)
-  PennTree Bank (Table 3 (right))
    - AWD + LSTM + TwinNet has valid perplexity: 61.0 (vs 61.2) and Test perplexity: 58.3 (vs 58.8)

The improvements in results for sequential MNIST and Wikitext-2 experiments show that TwinNet may also be effective for the case of unconditional generation. In MNIST, the use of a much larger regularization hyperparameter (1.5) was necessary. In PTB, the improvements are minor and more consistent in WikiText-2, which wasn’t included in our experiments. These experiments suggest that our method is applicable in for wider set of tasks comparing to the claim in the earlier version of our paper that TwinNet was better suited for conditional generation tasks.

---

### Decision · Program_Chairs · 2018-01-29
**ICLR 2018 Conference Acceptance Decision**

**Decision:**

Accept (Poster)

**Comment:**

Simple idea (which is a positive) to regularize RNNs, broad applicability, well-written paper. Initially, there were concerns about  comparisons, but he authors have provided additional experiments that have made the paper stronger.